# Exploring Gut Microenvironment in Colorectal Patient with Dual-Omics Platform: A Comparison with Adenomatous Polyp or Occult Blood

**DOI:** 10.3390/biomedicines10071741

**Published:** 2022-07-19

**Authors:** Po-Li Wei, Ming-Shun Wu, Chun-Kai Huang, Yi-Hsien Ho, Ching-Sheng Hung, Ying-Chin Lin, Mei-Fen Tsao, Jung-Chun Lin

**Affiliations:** 1Division of Colorectal Surgery, Department of Surgery, Taipei Medical University Hospital, Taipei Medical University, Taipei 110, Taiwan; poliwei@tmu.edu.tw; 2Cancer Research Center, Taipei Medical University Hospital, Taipei Medical University, Taipei 110, Taiwan; 3Translational Laboratory, Department of Medical Research, Taipei Medical University Hospital, Taipei Medical University, Taipei 110, Taiwan; 4Department of Surgery, College of Medicine, Taipei Medical University, Taipei 110, Taiwan; 5Graduate Institute of Cancer Biology and Drug Discovery, Taipei Medical University, Taipei 110, Taiwan; 6Division of Gastroenterology, Department of Internal Medicine, Wan Fang Hospital, Taipei Medical University, Taipei 116, Taiwan; mswu@tmu.edu.tw; 7Division of Gastroenterology and Hepatology, Department of Internal Medicine, School of Medicine, College of Medicine, Taipei Medical University, Taipei 110, Taiwan; 8International Master/Ph.D. Program in Medicine, College of Medicine, Taipei Medical University, Taipei 110, Taiwan; 9Integrative Therapy Center for Gastroenterologic Cancers, Wan Fang Hospital, Taipei Medical University, Taipei 116, Taiwan; 10Department of Laboratory Medicine, Wan Fang Hospital, Taipei Medical University, Taipei 116, Taiwan; 99421@w.tmu.edu.tw (C.-K.H.); 94445@w.tmu.edu.tw (Y.-H.H.); oryx@w.tmu.edu.tw (C.-S.H.); 11School of Medical Laboratory Science and Biotechnology, College of Medical Science and Technology, Taipei Medical University, Taipei 110, Taiwan; 12Ph.D. Program in Medical Biotechnology, College of Medical Science and Technology, Taipei Medical University, Taipei 110, Taiwan; 13Department of Family Medicine, School of Medicine, College of Medicine, Taipei Medical University, Taipei 110, Taiwan; greening1990@gmail.com; 14Department of Family Medicine, Wan Fang Hospital, Taipei Medical University, Taipei 116, Taiwan; 15Department of Medical Laboratory, Taipei Medical University Hospital, Taipei Medical University, Taipei 110, Taiwan; b8204021@tmu.edu.tw; 16Pulmonary Research Center, Wan Fang Hospital, Taipei Medical University, Taipei 116, Taiwan

**Keywords:** adenomatous polyp, colorectal cancer, gut microbiota, metabolite, Oxford Nanopore Technology

## Abstract

The gut mucosa is actively absorptive and functions as the physical barrier to separate the gut ecosystem from host. Gut microbiota-utilized or food-derived metabolites are closely relevant to the homeostasis of the gut epithelial cells. Recent studies widely suggested the carcinogenic impact of gut dysbiosis or altered metabolites on the development of colorectal cancer (CRC). In this study, liquid chromatography coupled-mass spectrometry and long-read sequencing was applied to identify gut metabolites and microbiomes with statistically discriminative abundance in CRC patients (*n* = 20) as compared to those of a healthy group (*n* = 60) ofenrolled participants diagnosed with adenomatous polyp (*n* = 67) or occult blood (*n* = 40). In total, alteration in the relative abundance of 90 operational taxonomic units (OTUs) and 45 metabolites were identified between recruited CRC patients and healthy participants. Among the candidates, the gradual increases in nine OTUs or eight metabolites were identified in healthy participants, patients diagnosed with occult blood and adenomatous polyp, and CRC patients. The random forest regression model constructed with five OTUs or four metabolites achieved a distinct classification potential to differentially discriminate the presence of CRC (area under the ROC curve (AUC) = 0.998 or 0.975) from the diagnosis of adenomatous polyp (AUC = 0.831 or 0.777), respectively. These results provide the validity of CRC-associated markers, including microbial communities and metabolomic profiles across healthy and related populations toward the early screening or diagnosis of CRC.

## 1. Introduction

Epidemiological studies indicated that colorectal cancer [CRC] is the third most common morbidity and has the second highest mortality rate among all cancers. With high mortality and increasing incidence, CRC is classified as the second leading cause of cancer-mediated deaths worldwide [1,2]. Morbidity and mortality of CRC is related to complex factors, including genetic susceptibility, dietary intake, or environmental stress [3,4]. Even though the correlation of CRC occurrence or progression with gut environment composed of microbiota and metabolites is increasingly acknowledged, demonstrating the influence of gut microbiomes on susceptibility to or development of CRC remains a critical challenge.

Emerging research has demonstrated the impact of specific bacteria or dysbiosis of resident microbiota on carcinogenesis of gut epithelium via activating inflammatory, mis-regulated cell proliferation, or altered genome stability [5]. The association of CRC development with an increase in the relative abundance of *Fusobacterium nucleatum* in gut microbiota has been widely identified [6]. Continuous studies demonstrated the influence of *Fusobacterium nucleatum* on activating Cdk5-mediated Wnt/β-catenin pathways, subsequently promoting CRC progression [7]. An increase in the relative abundance of *Peptostreptococcus anaerobius*, a Gram-positive anaerobic bacterium in the fecal or tissue samples of CRC patients was noted as well in recent reports [8]. *Peptostreptococcus anaerobius* exhibits the activity on activating the PCWBR2-integrin a2/b1-PI3K-Akt-NFkB axis throughout CRC occurrence, which was considered a potential therapeutic target toward CRC [9]. In contrast, the supplementation of probiotics, including *Bifidobacterium*, *Lactobacillus* genera, or *Clostridium butyricum*, was demonstrated to improve clinical prognosis or suppress development of CRC via lessening of CRC-associated signaling or manipulating gut microbiota [10,11,12].

Despite great effort being taken to comprehend the correlation of gut microbiota profiles with disease, the influence of gut microorganisms on homeostasis of host cells is primarily conveyed through metabolite-mediated pathways [13]. A diet-derived or artificial small molecule was identified as a signal to manipulate the risk toward the occurrence or progression of CRC [14]. Untargeted metabolomics is considered a well-established approach for the comprehensive and concomitant identification of metabolite composition that is increasingly applied to reveal the alteration in metabolism throughout the development of diverse diseases, including CRC [15]. Short-Chain Fatty Acids [SCFAs], including acetic acid, propionic acid, and butyric acid, are predominant metabolite generated by diverse bacteria with the intake of dietary fiber. Presence of SCFA was relevant to the permeabilization of lysosomal membrane or mitochondrial malfunction which subsequently sensitized the CRC cells toward programmed cell death [16]. *Clostridium butyricum* was documented to lessen the proliferation of CRC cells by manipulating Wnt/β-catenin signaling with the production of butyric acid [17]. In contrast, bile acid [BA] is first derived from cholesterol in the liver and further metabolized by gut microorganisms, including *Firmicutes*, *Bacteroidetes*, or *Actinobacteria* genera to genotoxic secondary BA [18,19]. Intake of high-fat diets resulted in the colonic excretion of secondary BA, such as deoxycholic acid and lithocholic acid, which diminishes the tumor-suppressive effect of the farnesoid X receptor signaling toward the carcinogenic process of CRC [20,21]. Nevertheless, the interplay between gut dysbiosis, gut metabolite composition, and gene expression profile throughout the development of CRC is largely uncharacterized.

In this study, the gut dysbiosis and fecal metabolite profile in CRC patients (*n* = 20) was classified using long-read sequencing and a LC-QTOFMS platform compared to those of patients diagnosed with colonic occult blood (*n* = 40), adenomatous polyp (*n* = 67), or healthy participants (*n* = 60). The increases in the relative abundance of *Peptostreptococcus stomatis*, *Shigella boydii*, allocholic acid, and S-Adenosylhomocysteine were specifically identified in feces samples of CRC patients. The results of a random forest regression model suggested that CRC-related microbial and metabolite composition have the potential to serve as an auxiliary test toward the early prediction of CRC occurrence. The impact of gut dysbiosis and metabolite-mediated mechanisms involved in CRC occurrence and development is worthy of further investigation.

## 2. Materials and Methods

### 2.1. Ethics Statement of Sample Collection

The procedure regarding recruitment of healthy participants and patients was approved by the Joint Institutional Review Board of Taipei Medical University (TMU; approval no. 201901013). CRC patients were recruited from the Division of Colorectal Surgery at Taipei Medical University. Patients diagnosed with adenomatous polyp were recruited from the Division of Gastroenterology at Taipei Municipal Wan Fang Hospital. The healthy participants and patients diagnosed with colonic occult blood were enrolled at the Department of Family Medicine at Taipei Municipal Wan Fang Hospital. The medical intervention, such as taking antibiotics, chemotherapy or radiation therapy, or taking feces softener for 3 months, were the exclusion criteria in this study.

### 2.2. Extraction of Bacterial Genomic DNA

Feces samples were collected using DNA/RNA Shield Fecal Collection tubes (Zymo Research, Irvine, CA, USA) to prevent contamination. Total genomic DNA were isolated from feces by using a Quick-DNA Fecal/Soil Microbe Microprep Kit (Zymo Research, Irvine, CA, USA) according to the manufacturer’s instructions. The concentration of the genomic DNA sample was quantified by using a fluorometric assay (GeneCopoeia, Rockville, MD, USA).

### 2.3. Sequencing of 16S Ribosomal RNA Gene

Microbial community in gut of the recruited participants was classified with utilization of a MinION sequencing platform (Oxford Nanopore Technologies (ONT), Oxford, UK). In total, 10 ng of total gDNA was subjected for library construction by using the SQK-16S024 Barcoding kit (ONT) according to the manufacturer’s protocol. The library was sequentially washed and then eluted from the magnetic beads (AMPure XP, Beckman Coulter, High Wycombe, UK). Then, 2 ng of barcoded DNA of each participant was pooled, adapter-ligated, and sequenced on a flow cell (FLO-MIN106D R9.4.1; ONT). The average read number of each sample was set to 100,000 to meet a reading depth of over 100.

### 2.4. Extraction of Fecal Metabolites

A total of 50 mg fece was mixed with 1 mL extract solution (acetonitrile: methanol: water = 2:2:1) and subjected to vigorous vortex. The mixture was homogenized, sonicated, and incubated at −20 °C for 1 h. After centrifugation (12,000× *g* rpm for 15 min at 4 °C), the supernatant was transferred to a glass vial.

### 2.5. UPLC-MS/MS Analysis

In total, 10 μL of supernatant was injected into a vanquish-focused ultra-high-performance liquid chromatography (UHPLC) system coupled with an Orbitrap Elite Mass Spectrometry (Thermo Fisher Scientific; San Jose, CA, USA). The binary mobile phase was composed of deionized water with 0.1% formic acid (solvent A) and LC-MS grade acetonitrile with 0.1% formic acid (solvent B). Blank injection was utilized to diminish the carry over effect prior to each injection. A QC injection was performed for normalization after every five injections. Mass spectrometry data were collected in positive mode with a default data-dependent acquisition method. An MS full1 scan was performed in profile mode at 60,000 resolution, followed by 10 data-dependent MS2 scans at 15,000 resolution. The mass scan range was set from 70 to 1000 m/z.

### 2.6. Processing, Annotation, and Statistical Analysis of Sequencing Results

The quantity of sequencing reads was accessed using Microbial Genomics Module (CLC genomics workbench (Qiagen v22.0.1; CLC bio, Aarhus, Denmark)). Qualified reads were mapped to the 16S rRNA reference released by the NCBI database. Annotation, taxonomic diversity, and relative abundance of microbial profiling were assessed using the Microbial Genomics Module (CLC genomics workbench (Qiagen v22.0.1). Relative alteration of identified taxa between each group was evaluated using the linear discriminant analysis (LDA) effect size (LEfSe) method with the default setting (https://huttenhower.sph.harvard.edu/galaxy/root (accessed on 23 March 2022)). The relative abundance of taxa was identified as significantly different with a *p* value < 0.05 and an LDA score (log10) > 3.

### 2.7. Processing and Annotation of UPLC-MS/MS Data

The original data was converted to the mzXML format using ProteoWizard software (V2.0, Taipei, Taiwan). Detection of signal strength, signal extraction, alignment, and integration of the original result was assessed using an XCMS-based program using R program. A BiotreeDB-based MS2 database was utilized for the annotation of metabolite profile with the cutoff value of 0.3.

### 2.8. Statistical Analysis

Statistics regarding the generated results were shown as the mean ± standard error. A one-way analysis of variance (ANOVA) combined with Tukey’s multiple comparison post-hoc test was used to compare continuous variables. A variable was identified significant with a *p* value of <0.05 (* *p* < 0.05; ** *p* < 0.01; *** *p* < 0.005). Zero-inflated negative binomial (ZINB) regression (R package pscl) was applied to assess the association between CRC-enriched metabolite and OTU. In brief, the read number of identified OTU was identified as a dependent variable and the identified strength of metabolite was considered a independent variable in the ZINB regressions. The association was shown by −log10(*p*-value)*sign (Beta) and the results of ZINB regressions and Beta presented the regression of the metabolite. The predictive utility of OTU or metabolite to the occurrence of CRC was estimated with the utilization of the receiver operating characteristic (ROC) curve and area under the ROC curve (AUC) ratio by using SPSS Statistics 19 (IBM, Armonk, NY, USA).

## 3. Results

### 3.1. Metadata of Enrolled Participants in This Study

In total, 20 CRC patients, 40 patients diagnosed with colonic occult blood, 67 patients diagnosed with colonic adenomatous polyp, and 60 healthy participants were recruited in this study. No statistical difference in the included confounders of age, gender, regular exercise, or a history of smoking or drinking was noted among all groups (Table 1, *p* > 0.05). Nevertheless, the relevance of CRC occurrence with other primary malignancy (*p* < 0.01) or family history of cancer (*p* < 0.01) was noted in this study.

### 3.2. Statistical Analysis of Sequencing Throughput in Each Enrolled Group

For characterizing the gut microbial communities of enrolled participants in this study, the genomic DNA extracted from the fecal sample was subjected to the long-read sequencing platform (MinION, ONT, Oxford, UK). Numbers with an average of sequenced and qualified reads per sample were evaluated by using the CLC Genomics Workbench software (v.22.0.1; Aarhus, Denmark). As shown in Table 2, no statistical difference was noted regarding the sequencing efficiency with the DNA extracted from each group (Table 2).

Statistical difference in species diversity (α-diversity) between the microbial communities of healthy participants and CRC patients (*p* < 0.05) or patients diagnosed with colonic occult blood (*p* < 0.005), but not patients diagnosed with adenomatous polyp (*p* > 0.05), was identified with the analyses using the Simpson index (Figure 1A) or Shannon entropy (Figure 1B). The dissimilarity between the microbial communities in the distinct group was subsequently evaluated with the Weighted Unifract distance (Figure 2A) or Bray-Curtis index (Figure 2B). Statistical results of the principal coordinate analyses (PCoA) showed the unique population aggregates in fecal samples of CRC patients (Figure 2, red dot), patients diagnosed with adenomatous polyp (Figure 2, orange dot), or patients diagnosed with colonic occult blood (Figure 2, green dot), compared to those of the healthy group (Figure 2, blue dot), which constituted the disease or disorder-related enterotype [22].

### 3.3. Identification of CRC-Associated Microbial Community with Result of Long-Read Sequencing

The long-read sequencing approach has been reported to exhibit a resolution for taxonomic identification of the microbial community at the species level [23]. In this study, around 1000 operational taxonomic units (OTUs) were identified in individual groups (Table 2). The bar chart presented top 25-ranked OTUs based on the average reads classified from all recruited group with the utilization of MinION sequencing results coupled with the CLC Genomics Workbench (Qiagen) pipeline (Figure 3 and Appendix A). A heat map was generated based on 19 OTUs with relatively high abundances and 13 OTUs with relatively low abundance and statistically significance in CRC group as compared to the healthy participants (Appendix A; *p* < 0.05; FDR < 0.05; Bonferroni < 0.05). The results illustrated the relevance of identified CRC-enriched OTUs from all enrolled participants. Among these candidates, the gradual increases in the relative abundant levels of 9 OTUs (Figure 4, red character) in gut microbial communities were noted from the enrolled patients diagnosed with colonic occult blood (Figure 4, green), adenomatous polyp (Figure 4, orange) to CRC patients (Figure 4, red). In contrast, the decreases in relative abundances of 13 OTUs was identified in enrolled CRC patients as compared to those of healthy participants (Figure 4, blue character). The adenomatous polyp-associated abundance of *Pectobacterium* or *Klebsiella* genera was noted in a minority of CRC patients (Appendix A, red character). Nevertheless, the relevance of colonic occult blood-related microbial communities with occurrence of adenomatous polyp or CRC was sparsely identified in this study (Appendix A). A linear discriminant analysis (LDA) effect-size (LEfSe) assay was subjected to further evaluate the differential abundances of identified OTUs between healthy participants, patients with distinct disorders, and CRC patients. Discriminative abundance was considered statistically convincing for *p* value of <0.05 and a logarithmic LDA score cutoff of >3 or <−3. The statistical score showed the numerically abundant *Peptostreptococcus stomatis*, *Escherichia marmotae*, *Brenneria alni*, *Shigella boydii*, and *Trabulsiella odontotermitis* in the microbial communities of CRC patients (Figure 5a, red bar) compared to the healthy group (LDA score (log 10) < −3). In contrast, the relative abundance of *Prevotella copri*, *Selenomonas ruminantium,* and *Coprococcus comes* were much higher in the gut microbial communities of healthy participants than those of enrolled CRC patients (LDA score (Figure 5a, green bar; LDA score (log 10) > 3). The statistical results showed gradual increases in the relative abundance of *Peptostreptococcus stomatis*, *Escherichia marmotae*, *Brenneria alni*, *Shigella boydii,* and *Trabulsiella odontotermitis* in gut microbial communities of the enrolled patients diagnosed with colonic occult blood, adenomatous polyp, and CRC patients as compared to those of healthy participants (Figure 5b).

### 3.4. Untargeted Identification of CRC-Associated Metabolite Profile with LC-MS/MS Analysis

With the utilization of the UPLC-MS/MS platform and coupling analytic pipeline, a total of 187 metabolites were quantified in the fecal samples of enrolled patients. The result of the principal component analysis (PCA) showed the statistical difference in the gut metabolite profiles among CRC patients (Figure 6, red dot), patients with adenomatous polyp (brown dot) or colonic occult blood (green dot), and healthy subjects (blue dot) (PERMANOVA, *p* = 0.001). As shown in Appendix A, a total of 45 metabolites with discriminating abundance between healthy participants and CRC patients were identified with the criteria, including a variable importance in projection value (VIP) > 1.5, a significant alteration in relative abundance (−2 > fold-change > 2), and a significant *p* value < 0.05. As shown in Figure 7, a heat map was shown to illustrate the relatively differential abundances of identified metabolites among the enrolled groups as compared to those of healthy group with a significant alteration (−2 > fold-change > 2) and statistical result (*p* value < 0.05; FDR value < 0.05). Among the candidates, the gradual increases in the relative abundances of four metabolites, including S-Adenosylhomocysteine, N,N-Dimethylaniline, Stearic acid, and allocholic acid, were noted in gut microbial communities of the enrolled CRC patients as compared to those of healthy participants (Figure 7). These results suggested the potential application of identified metabolites to function as the disorder or CRC-associated biomarker.

Subsequently, the Zero-inflated negative binomial (ZINB) regression (R package pscl) was applied to evaluate the relevance between CRC-enriched metabolites and 19 OTUs with relatively high abundance at the species level in CRC patients [24]. Among the identified OTUs discriminating the enrolled CRC patients from healthy participants, the statistically significant association between *Shigella*, *Escherichia*, *Enterobacter*, *Streptococcus,* and *Peptostreptococcus* and six CRC-enriched metabolites, including L-Phenylalanine, L-valine, Gibberellin A3, S-Adenosyl Homocysteine, stearic acid, and allocholic acid was classified along with CRC occurrence (Figure 8, *p* < 0.05). These results indicated the significant associations among particular gut metabolites and OTU which were relevant to colorectal carcinogenesis.

### 3.5. Predictive Utility of Identified Gut OTUs or Metabolites toward Occurrence of Adenoma or CRC

To estimate the utility of identified OTU or metabolite on distinguishing patients with CRC or adenomatous polyp from healthy participants, a random forest regression model was applied with the relative abundances of identified OTUs or metabolite using the receiver operating characteristics (ROC) curve. The ROC curve were generated with the relative abundance of five identified OTUs, including *Peptostreptococcus stomatis*, *Trabulsiella odontotermitis*, *Shigella boydii*, *Brenneria alni*, and *Escherichia marmotae*, or four CRC-enriched metabolites, such as S-Adenosylhomocysteine, N,N-Dimethylaniline, stearic acid, and allocholic acid. The same identified OTUs discriminated adenomas occurrence from the healthy group with an area under the ROC curve (AUC) of 0.713 and CRC occurrence from healthy participants with and AUC of 0.785 (Figure 9, left). To distinguish adenoma from healthy groups, four CRC-enriched metabolites were characterized with an AUC of 0.725 and CRC occurrence from healthy condition with an AUC of 0.756 (Figure 9, middle). To estimate whether more significant discrimination between healthy conditions and CRC or adenoma occurrence was achieved, the CRC-enriched OTUs and metabolites with significant association as demonstrated in Figure 8 were subjected to the ROC assay. For distinguishing CRC from the healthy group, utilization of five CRC-enriched metabolites (L-Phenylalanine, Gibberellin A3, S-Adenosyl Homocysteine, stearic acid, and allocholic acid) and four CRC-related OTUs (*Peptostreptococcus stomatis*, *Shigella boydii*, *Enterobacter hormaechei,* and *Streptococcus lutetiensis*) resulted in a higher AUC of 0.9155 (Figure 9, right) as compared to that with only metabolites or OTUs. The same combination exerted better discrimination to adenoma from the healthy group with an AUC of 0.9027 (Figure 9, right) than that with only metabolites or OTUs. These results demonstrated that the relevant gut dysbiosis-associated metabolite constituted a potential biomarker to the diagnosis of adenoma or CRC occurrence.

## 4. Discussion

Along with the advance of the analytic pipeline for microbial and metabolomic profiling, accumulating results continuously illustrate the impact of gut microbiota and metabolites on colorectal tumorigenesis. Emerging perspectives, including early prevention, screening, or diagnosis of CRC might be applied with clinical treatment [25]. It is crucial as well to realize whether gut dysbiosis or toxic metabolite is causative of CRC occurrence or progression, or consequent biomarkers of the disease state. Herein, we executed a cross-sectional cohort study to identify microbial communities and metabolomics profile that functioned as putative markers toward diagnosis or prediction of CRC.

Immunological fecal occult blood test [iFOBT] has been a non-invasive method and continuously implemented for early screening in average-risk populations of CRC worldwide [26]. Nevertheless, the highly diagnostic sensitivity of iFOBT toward CRC comes at the cost of false positive rate [27]. It has been widely documented the potential of fecal bacteria on serving an auxiliary biomarker for non-invasive screening in average or high-risk population of CRC, such as *Fusobacterium*, *Escherichia*, or *Bacteroides* genera [28,29]. Nevertheless, diverse variation was noted among wide studies for the identification of CRC-associated microbial markers [30]. Insufficient reading depth of taxonomy and high dependence on reference database resulted in the diverse variation regarding the identification of CRC-related OTU [31]. Several pioneering studies focused on the discriminating microbial communities in CRC patients from the healthy counterparts by executing the cross-section cohort assays [32]. Other studies aimed at identifying the microbial communities associated with diagnosis of adenomatous polyp to serve the early screening biomarker toward CRC occurrence or progression [33]. The execution of a longitudinal cohort study constitutes a practicable strategy to shed light on the relevance of gut dysbiosis with the progression of adenomatous polyp to CRC occurrence. The advancement of long-read sequencing approach conferred species-level resolution toward the identification of gut microbiota, and resulted in the construction of an emerging and convincing model with an identified feature to discriminate CRC from adenoma or a healthy state [34]. In addition to a 2.64-fold increase in the relative abundance of *Fusobacterium nucleatum*, the relatively high levels of emerging species, such as *Trabulsiella*, *Franconibacter*, and *Brenneria* genera in gut microbial communities of CRC patients compared to those of healthy participants was noted in this study. Nevertheless, the impact or relevance of this phenomenon to pathogenesis of CRC should be further pursued with larger CRC patient cohorts.

Accumulating reports suggested that gut metabolomic profiling represents another promising approach for identifying CRC-specific biomarkers [35]. Aberrant abundances of particular amino acids and the derived metabolites generated by particular bacteria were widely characterized in the fecal samples of CRC patients as compared to the healthy counterparts, which were relevant to the active proliferation of CRC cells by modulating immune response or epigenetic regulation [36,37]. A high-fat diet was demonstrated to result in colonic excretion of secondary bile acids (BAs), such as deoxycholic acid and lithocholic acid, which are metabolized by specific gut flora, including *Firmicutes*, *Bacteroidetes*, and *Actinobacteria* genera [38,39]. Accumulation of secondary BAs was documented to mediate the aberrant activation of Wnt/β-catenin, TGR5, or EGFR signaling pathways, which was relevant to immortality, impaired intestinal barrier, or poor prognosis of colonic carcinogenesis [7,21,40]. In contrast, supplementation of dietary fiber led to increases in the gut short-chain fatty acids (SCFAs), including acetic acid, propionic acid, and butyric acid, which are metabolized by *Faecalibaculum rodentium* and *Clostridium butyricum* [41,42]. Published reports documented that combined supplementation of diverse SCFAs or SCFA-generated flora diminished tumor formation or lessened colonic inflammation in a mouse model [43,44]. In addition to the identification of CRC-enriched metabolites in fecal samples, the metabolite-species association was assessed by using the ZINB regression model. Among the interplays, the relevance of *Streptococcus* genera or other species with conversion or metabolism of S-Adenosylhomocysteine was previously reported [45]. Relatively high levels of S-Adenosylhomocysteine, a metabolite involved in methionine metabolism and methylated modification, was identified in cancerous mucosa as compared to adjacent normal tissues of CRC patients [46]. An elevation of S-Adenosylhomocysteine was reported to result in hypermethylation of the promoter region and an impaired antioxidant mechanism, which was closely related to oncogenesis [47]. These results suggested the potential causation or correlation of metabolite-species interplay with the pathogenesis of CRC. Nevertheless, the absence of standardization in identifying procedure and analytic pipeline regarding metabolomic profiling is the limitation and critical issue for defining gut metabolites as a clinical biomarker for clinical screening, prediction, or diagnosis.

## 5. Conclusions

In this cross-section cohort study, the relatively high-level structures of gut microbiota or metabolites in CRC patients different from other counterparts was identified using a dual-omics approach composed of a long-read sequencer and UPLC-MS/MS platform. The identified fecal microbiota or metabolite profiling exhibited a distinct utility in differentiating the patients diagnosed with CRC from other enrolled participants. A longitudinal study was subsequently conducted by employing the microbiota or metabolite-based candidate to evaluate the participant for the risk of being diagnosed with CRC. The detailed functional analysis regarding the impact of CRC-associated microbiota or metabolite community is crucial as well for driving the transformation of gut environment-derived strategies into precision screening and diagnosis of CRC.

## Figures and Tables

**Figure 1 biomedicines-10-01741-f001:**
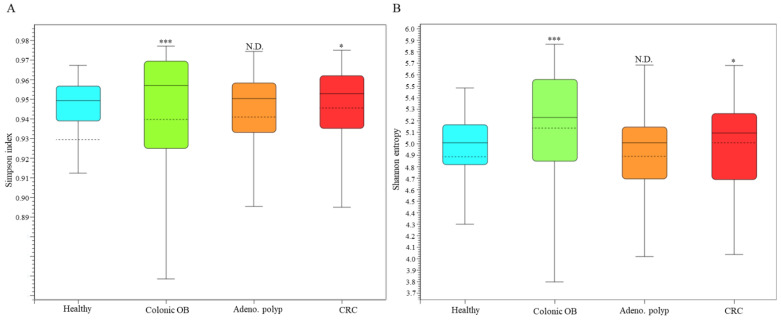
Diversity of taxonomic alignments between healthy group (blue), Colonic OB (green), Adenomatous polyp (brown), and CRC (red) with long-read sequencing results. The α-diversity in all groups is illustrated using (**A**) Simpson index and (**B**) Shannon entropy (No difference (N.D.) > 0.05; * *p* < 0.05; *** *p* < 0.005).

**Figure 2 biomedicines-10-01741-f002:**
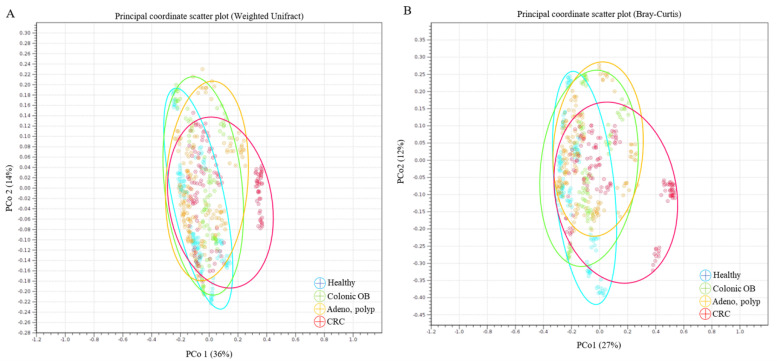
The dissimilarity of gut microbial community among the enrolled participants with sequencing results is identified using principal component analysis (PCoA), including (**A**) Weighted Unifrac and (**B**) Bray-Curtis method.

**Figure 3 biomedicines-10-01741-f003:**
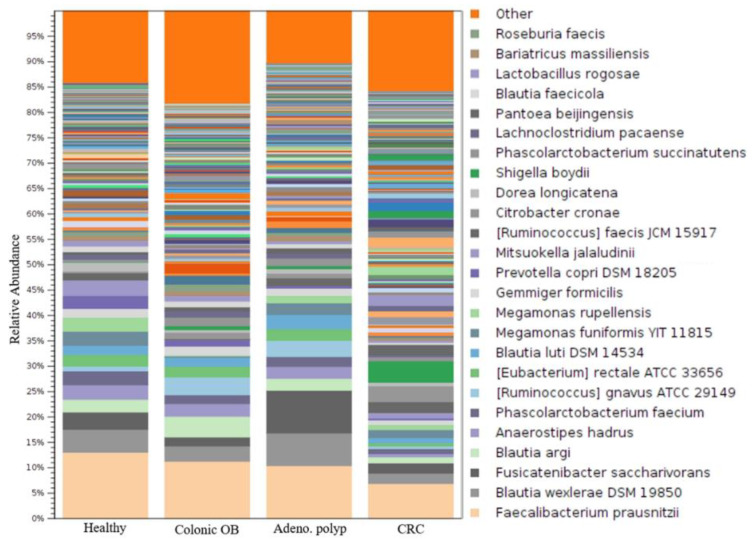
Identification of operational taxonomy unit (OTU) in healthy participants and enrolled patients with MinION sequencing results. Stacked bar chart is applied to present the relative abundances of the top 25 classified OTUs to species level.

**Figure 4 biomedicines-10-01741-f004:**
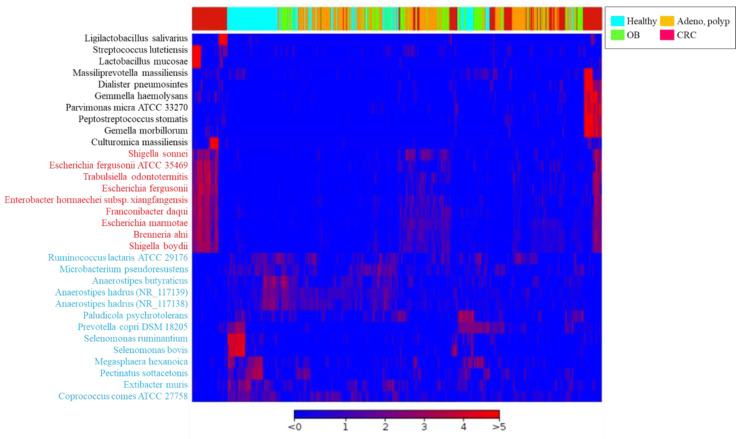
The relevance of 19 CRC-enriched OTUs (black and red character) and 13 OTUs with relatively low abundances in CRC patients as compared to those of the healthy participants (blue character) at the species level among all recruited participants is illustrated using a heatmap chart.

**Figure 5 biomedicines-10-01741-f005:**
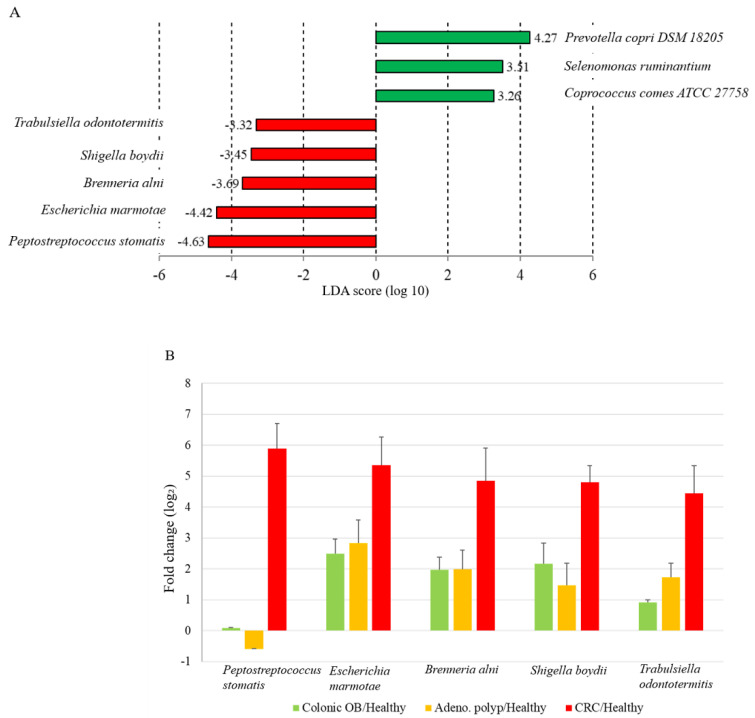
Differential abundances of identified OTU at the species level between healthy participants and enrolled patients. (**A**) Histogram of linear discriminant analysis (LDA) scores presents differential abundances of identified OTUs in healthy participants (green bar) and CRC patients (red bar). (**B**) Relative abundances of identified OTUs in the fecal samples of enrolled patients diagnosed with colonic occult blood, adenomatous polyp, and CRC.

**Figure 6 biomedicines-10-01741-f006:**
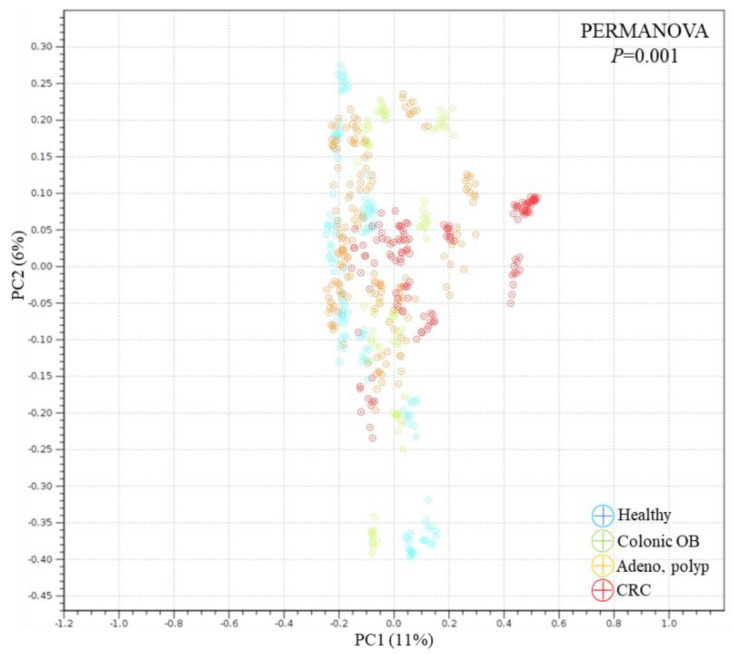
A Principal component analysis (PCA) is applied to estimate the dissimilarity of gut metabolomic profiling between healthy participants and enrolled patients diagnosed with colonic occult blood, adenomatous polyp, and CRC.

**Figure 7 biomedicines-10-01741-f007:**
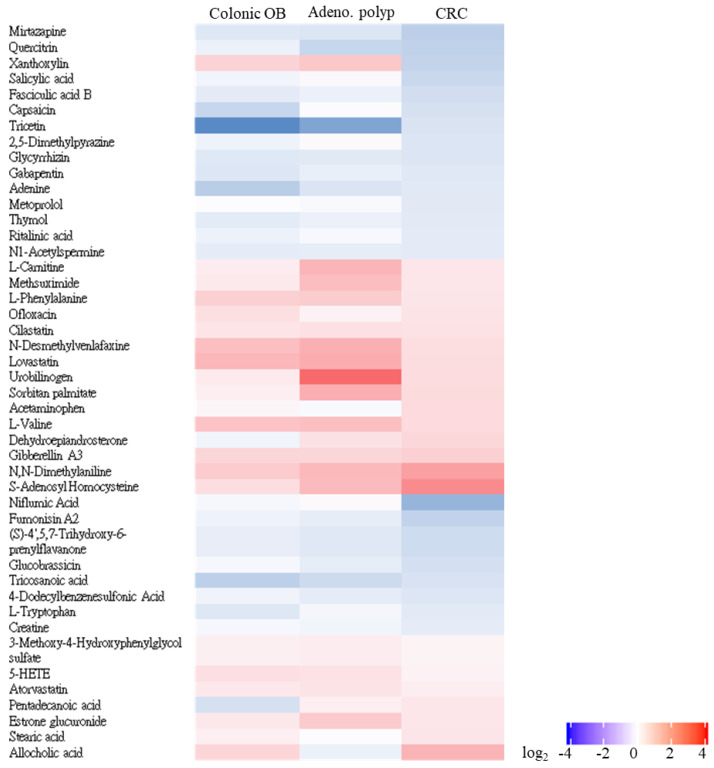
Z-score heatmap is constructed with 45 distinctly differential metabolites between enrolled patients diagnosed with colonic occult blood, adenomatous polyp, and CRC. Significance of identified metabolites were evaluated using variable importance in projection value (VIP) and alteration in relative abundance from pairwise PLD-DA analysis and Wilcoxon rank-sum test, with VIP > 1.5, alteration in relative abundance (−2 > fold-change > 2), *p* value < 0.05, and FDR value < 0.05 as the cut-off for significance.

**Figure 8 biomedicines-10-01741-f008:**
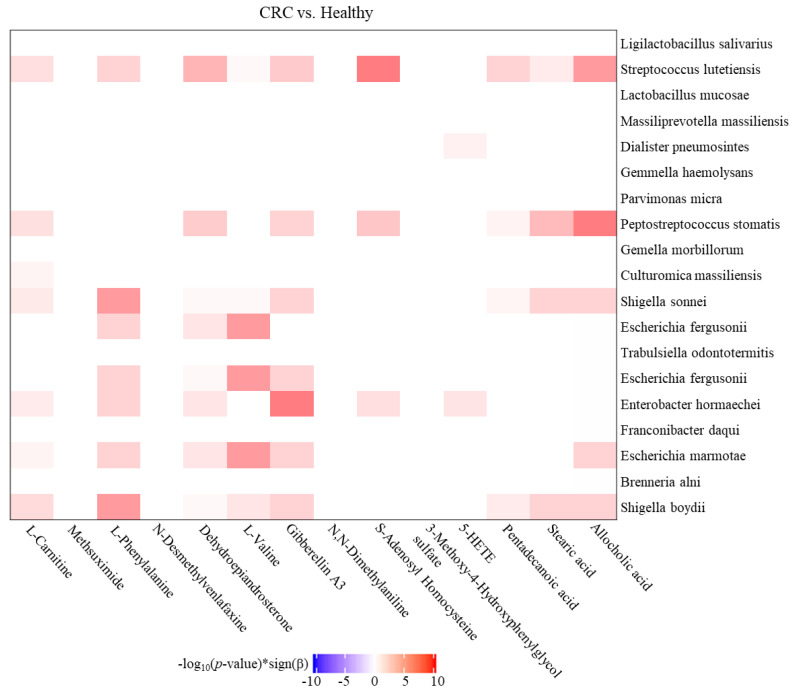
Associations among CRC-enriched metabolites and gut dysbiosis in enrolled CRC patients. Heatmap for the relevance between metabolites and OTUs along with CRC occurrence. The metabolites-OTUs associations were evaluated by using zero-inflated negative binomial (ZINB) regressions. The strengths of associations were measured by -log10 (*p*-value)*sign (Beta) from the results of ZINB regressions and *p* value < 0.05 was identified as the cut-off for significance.

**Figure 9 biomedicines-10-01741-f009:**
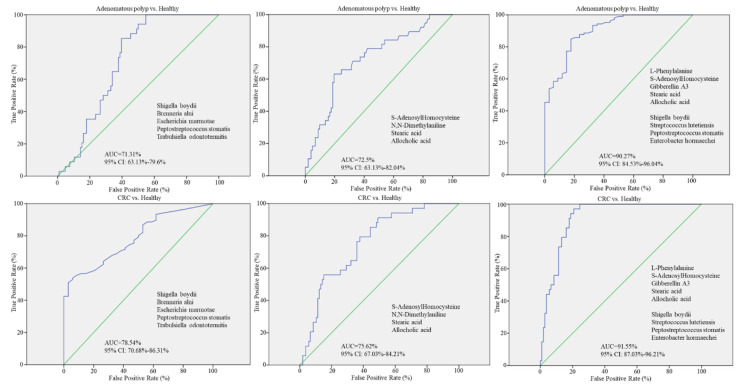
Predictive utility of identified gut OTUs or metabolites toward the occurrence of CRC or adenomatous polyp was evaluated using the random forests model. The area under the receiver operating characteristics (ROC) curve (AUC) was applied for differentiating CRC patients or enrolled patients diagnosed with adenomatous polyp from the healthy group with the relative abundances of identified OTUs (**left**), the intensity of identified gut metabolites (**middle**), or combination of gut dysbiosis-associated metabolites in CRC patients (**right**).

**Table 1 biomedicines-10-01741-t001:** Demographics of healthy participants and enrolled patients diagnosed with colonic occult blood, adenomatous polyp, and CRC.

Group	Healthy (*n* = 60)	Colonic OB (*n* = 40)	Adenomatous Polyp (*n* = 67)	CRC (*n* = 20)	*p*
Age (Median(IQR))	61 (31–72)	52 (35–63)	48 (39–60)	64 (43–88)	>0.05
Sex (*n*,%)					>0.05
Female	35 (58.33)	22 (55)	39 (58.2)	13 (65)	
Male	26 (41.67)	18 (45)	28 (41.8)	7 (35)	
History of cancer (*n*,%)	6 (11.32)	3 (8.33)	5 (11.63)	4 (20) (*p* < 0.01)	>0.05
Family history of cancer (*n*,%)	10 (16.67)	8 (20)	13 (19.4)	8 (40) (*p* < 0.01)	>0.05
History of smoking (*n*,%)	15 (25)	8 (20)	13 (19.4)	5 (25)	>0.05
History of drinking (*n*,%)	6 (10)	8 (20)	12 (17.91)	5 (20)	>0.05
History of regular exercise (*n*,%)	27 (45)	21 (52.5)	24 (35.82)	8 (40)	>0.05

**Table 2 biomedicines-10-01741-t002:** Statistical summary of long-read sequencing results.

Group	Healthy (*n* = 60)	Colonic OB (*n* = 40)	Adenomatous Polyp (*n* = 67)	CRC (*n* = 20)	*p*
Number of Raw reads per sample	84,534 (±5079)	87,817 (±4121)	81,775 (±2719)	83,756 (±3217)	>0.05
Number of qualified reads per sample	62,749 (±3226)	65,292 (±2884)	51,944 (±2431)	54,645 (±2005)	>0.05
Reads in identified taxa	58,505 (±2845)	60,297 (±2355)	45,403 (±1977)	49,446 (±2105)	>0.05
Correctly classified (% (SD))	93.24 (±3.64)	92.35 (±4.97)	91.59 (±3.55)	90.49 (±2.69)	>0.05
Number of identified taxa per sample	1114	1075	931	948	>0.05

## Data Availability

The data presented in this study are available on request from the corresponding author due to privacy restrictions.

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
