# Peer review of "Exploring Gut Microenvironment in Colorectal Patient with Dual-Omics Platform: A Comparison with Adenomatous Polyp or Occult Blood"

_biomedicines, 2022, doi:10.3390/biomedicines10071741_

Round 1
Reviewer 1 Report
This manuscript explores the gut microbiome and metabolome using fecal samples to identify associations with CRC and certain subtypes. Unfortunately there have been multiple studies that have done similar results with larger patient cohorts that will affect the novelty of the study. Importantly these studies were not cited or discussed. Furthermore, there are some significant errors or incorrect data analysis performed in this study.
Main comments are:
Figure 1: What does the coloured dots in the box plot mean? There are more dots than the number of patients described (i.e. CRC seems to have > 20 samples). Does colour signify technical replicates? If so, the author should indicate this in the figure legend or manuscript. Furthermore, technical variation between replicates need to be discussed.
Figure 2B & Figure 6 appear to have the same plot even though the labels are different. How is this possible when one is measuring metabolites and the other microbiome independently? It’s likely that the same graph or data was used twice. Either is likely as the x-y information has changed. I would highly doubt the integrity of this manuscript if this was correct.
Figure 3 needs to clarify what the top 25-ranked OTUs based on. Is the top 25 based on the average reads of a species in a specific group? (i.e., Top 25 in healthy) or is this based on top 25 hits of species in the average from all groups? The authors should include an excel of the bacteria species in relative abundance, so that readers can appreciate the variation within each sample and each sample group.
In Figure 4, the authors stated that there is specific bacteria species enriched for CRC and healthy patients. The figure legend needs to be worded properly and the heatmap re-analysed. As is, this figure is confusing as red and blue is used multiple times (for expression and for identification of patient cohors). Hierarchal clustering doesn’t appear to be performed in this analysis, there are 2 CRC clusters found in this heatmap located at both edges of the heatmap. Furthermore, the healthy associated microbiome is not consistent throughout the healthy cohort, there appears to be 3 main bacterial patterns/groups that were clumped to “healthy associated”. It is not clear how 32 OTUs were selected. Even in the non-CRC enriched cohort (black) like D pneumosintes, clearly this associated with CRC.
In Figure 5: no statistical analysis was performed in these figures.
Figure 7 – Is the p-value adjusted for FDR? Furthermore, they need to explicitly say that the heatmap values are against healthy controls.
The authors performed dual-omics analysis and yet did not perform any analysis that incorporates both. The ROC curves performed were simplistic only studying the effect of one parameter, either 1 bacteria or metabolite. The ROC curves also don’t look correct, 95% CI should be included. There are published manuscripts that were able to combine the contributions of microbes/metabolites and this study will benefit from similar analysis.
The discussion needs to be expanded. In particular, discussions identifying any links between bacteria and metabolites will be important. How the findings contribute to disease progression and pathology should also be included.
Critically, assessment on how this data set compares with other similar studies which have done larger cohorts should be compared.
In CRC, gut microbe and metabolomic interegration has been studied in 2 recent examples:
1. https://doi.org/10.1186/s40168-021-01208-5, which studied fecal microbiome and metabolome
2. doi: 10.1136/gutjnl-2020-323476, fecal microbiome and serum metabolome
Author Response
Response to Reviewers' Comments
(Manuscript Number: biomedicines-1759887)
Dear Editor:
We thank you for your response and for allowing revision of our manuscript (biomedicines-1759887; Exploring gut microenvironment in colorectal patient with dual-omics platform: A comparison with adenomatous polyp or occult blood). The manuscript was revised in line with the suggestions and comments of all reviewers. We hope that the revised manuscript achieves reviewer satisfaction. Our point-by-point responses to all specific reviewer comments, suggestions, and queries are as follows.
REVIEWER(S)' COMMENTS
Reviewer #1: This manuscript explores the gut microbiome and metabolome using fecal samples to identify associations with CRC and certain subtypes. Unfortunately there have been multiple studies that have done similar results with larger patient cohorts that will affect the novelty of the study. Importantly these studies were not cited or discussed. Furthermore, there are some significant errors or incorrect data analysis performed in this study.
Response to Reviewer Comments
- Figure 1: What does the coloured dots in the box plot mean? There are more dots than the number of patients described (i.e. CRC seems to have > 20 samples). Does colour signify technical replicates? If so, the author should indicate this in the figure legend or manuscript. Furthermore, technical variation between replicates need to be discussed.
Response: In the previous version of Figure 1, a single dot indicates the microbial community classified with 4,000 sequencing reads generated with the DNA sample of each participant. More dots than the number of patients were therefore presented in the previous figure. Nevertheless, the presentation was not commonly applied in relevant study. To clearly illustrate the results, the color dots were omitted in the revised Figure 1.
- Figure 2B & Figure 6 appear to have the same plot even though the labels are different. How is this possible when one is measuring metabolites and the other microbiome independently? It’s likely that the same graph or data was used twice. Either is likely as the x-y information has changed. I would highly doubt the integrity of this manuscript if this was correct.
Response: I do appreciate your reminder and apologize for my great negligence for supervising the preparation of manuscript. All reported results, including Figure 2 and 6, throughout the revised manuscript were carefully checked and corrected.
- Figure 3 needs to clarify what the top 25-ranked OTUs based on. Is the top 25 based on the average reads of a species in a specific group? (i.e., Top 25 in healthy) or is this based on top 25 hits of species in the average from all groups? The authors should include an excel of the bacteria species in relative abundance, so that readers can appreciate the variation within each sample and each sample group.
Response: The bar chart presented top 25-ranked OTUs based on the average reads which were classified from all recruited group. The result was further described in the revised manuscript (please see page 6, lines 223-226). An excel file including the relative abundance of identified OTUs from all groups was provided as the Supplementary Table S1 in the revised manuscript.
- In Figure 4, the authors stated that there is specific bacteria species enriched for CRC and healthy patients. The figure legend needs to be worded properly and the heatmap re-analysed. As is, this figure is confusing as red and blue is used multiple times (for expression and for identification of patient cohors). Hierarchal clustering doesn’t appear to be performed in this analysis, there are 2 CRC clusters found in this heatmap located at both edges of the heatmap. Furthermore, the healthy associated microbiome is not consistent throughout the healthy cohort, there appears to be 3 main bacterial patterns/groups that were clumped to “healthy associated”. It is not clear how 32 OTUs were selected. Even in the non-CRC enriched cohort (black) like D pneumosintes, clearly this associated with CRC.
Response: The figure legend and the results were revised to properly illustrated the analytic results according to the reviewer's comment (please see Figure legend of Figure 4; page 6-7, lines 226-235). 19 OTUs with relatively high abundances and 13 OTUs with relatively low abundance and statistically significance (Supplementary Table S2; p<0.05; FDR<0.05; Bonferroni<0.05) in CRC group as compared to the healthy group were included for construction of the heatmap. The hierarchical clustering of heatmap was constructed based on the phylogenic correlation of identified OTUs among all group by using CLC Genomics Workbench software, consequently leading to multiple clusters of CRC patients, healthy participants, or other groups. Nevertheless, several CRC-enriched OTUs, such as D. pneumosintes, was not commonly classified in CRC patients, which suggested its inefficient utility to the prediction of CRC occurrence.
- In Figure 5: no statistical analysis was performed in these figures.
Response: The adjusted p-value and LDA-score cutoff regarding the LEfSe assay has been illustrated in the previous Materials and method section 2.6 (please see page 4, lines 161-162). within the previous manuscript. Nevertheless, the statistical results of LEfSe assay was further described in the revised manuscript (please see page 7, lines 242-243).
- Figure 7 – Is the p-value adjusted for FDR? Furthermore, they need to explicitly say that the heatmap values are against healthy controls.
Response: The result was further illustrated according to the reviewer's comment. The p-value of analytic results presented in Figure 7 was adjusted for FDR (please see page 8, lines 268-272; Legend of Figure 7).
- The authors performed dual-omics analysis and yet did not perform any analysis that incorporates both. The ROC curves performed were simplistic only studying the effect of one parameter, either 1 bacteria or metabolite. The ROC curves also don’t look correct, 95% CI should be included. There are published manuscripts that were able to combine the contributions of microbes/metabolites and this study will benefit from similar analysis.
Response: The metabolite-species association in CRC patients was evaluated by using the Zero-inflated negative binomial (ZINB) regression (R package pscl) (please see Figure 8; page 11, lines 306-314). The relative abundances of CRC-enriched OTUs or metabolites were applied to generate the ROC curve containing AUC value and 95% CI, which evaluated the predictive utility of gut dysbiosis or altered metabolite profile toward the occurrence of CRC or adenomatous polyp according to the reviewer's comment (please see Figure 9; page 11-12, lines 326-344).
- The discussion needs to be expanded. In particular, discussions identifying any links between bacteria and metabolites will be important. How the findings contribute to disease progression and pathology should also be included. Critically, assessment on how this data set compares with other similar studies which have done larger cohorts should be compared.
Response: The revised discussion regarding the metabolite-species correlation and its impact on pathogenesis of CRC was provided. Moreover, the potential impact of metabolite-species association that we identified on pathogenesis of CRC was discussed in the revised manuscript (please see Discussion section; page 13, lines 380-384; lines 385-413).
Reviewer 2 Report
In this study, the gut dysbiosis and fecal metabolite profile in CRC patients (n=20) was classified using long-read sequencing and the LC-QTOFMS platform compared to those of patients diagnosed with colonic occult blood (n=40), adenomatous polyp (n=67), or healthy participants (n=60). The increases in the relative abundance of Peptostreptococcus stomatis, Shigella boydii, Allocholic acid, and S-Adenosylhomocysteine were specifically identified in feces samples of CRC patients. Based on the results of a random forest regression model, the authors suggest that CRC-related microbial and metabolite composition has potential to serve an auxiliary test toward the early prediction of CRC occurrence.
In general, this is an interesting study employing modern techniques to investigate the microbial inhabitants and metabolites specific for CRC.
Main concerns/questions:
- Did all patients and controls undergo colonoscopy?
- Was CRC excluded in healthy patients?
- How did tumor stage, size or metastasis influence the microbial population?
- Were patients staged by CT-scan? Were the microbial results correlated with tumor stage or tumor localization? Right-sided vs left sided? Did age and MSI (microsatellite-instability) influence the microbial pattern?
- How were the adenomas compared to each other? What classification was used to stage them?
- How did diabetes etc impact the type and abundance of microbial inhabitants?
- Were surgical patients receiving bowel preparation or were they treated by single shot antibiotics? if yes, please discuss.
- The results should be discussed with more similar studies on this topic.
Author Response
Response to Reviewers' Comments
(Manuscript Number: biomedicines-1759887)
Dear Editor:
We thank you for your response and for allowing revision of our manuscript (biomedicines-1759887; Exploring gut microenvironment in colorectal patient with dual-omics platform: A comparison with adenomatous polyp or occult blood). The manuscript was revised in line with the suggestions and comments of all reviewers. We hope that the revised manuscript achieves reviewer satisfaction. Our point-by-point responses to all specific reviewer comments, suggestions, and queries are as follows.
REVIEWER(S)' COMMENTS
Reviewer #2: In this study, the gut dysbiosis and fecal metabolite profile in CRC patients (n=20) was classified using long-read sequencing and the LC-QTOFMS platform compared to those of patients diagnosed with colonic occult blood (n=40), adenomatous polyp (n=67), or healthy participants (n=60). The increases in the relative abundance of Peptostreptococcus stomatis, Shigella boydii, Allocholic acid, and S-Adenosylhomocysteine were specifically identified in feces samples of CRC patients. Based on the results of a random forest regression model, the authors suggest that CRC-related microbial and metabolite composition has potential to serve an auxiliary test toward the early prediction of CRC occurrence.
In general, this is an interesting study employing modern techniques to investigate the microbial inhabitants and metabolites specific for CRC.
Main concerns/questions:
Response to Reviewer Comments
- Did all patients and controls undergo colonoscopy?
Response:
All enrolled healthy participants and patients have underwent colonoscopy prior to the collection of feces.
- Was CRC excluded in healthy patients?
Response:
All CRC patients were excluded in the other groups in this study.
- How did tumor stage, size or metastasis influence the microbial population?
Response: I agree with the reviewer that the relevance between cancerous signature and gut dysbiosis is an interesting and important issue. Nevertheless, our results were not convincing to illustrate this issue with small CRC patient cohorts in this study.
- Were patients staged by CT-scan? Were the microbial results correlated with tumor stage or tumor localization? Right-sided vs left sided? Did age and MSI (microsatellite-instability) influence the microbial pattern?
Response: All CRC patients were staged with the CT-scan results and pathological examination results of cancerous tissues. Our results were not convincing to evaluate the relevance between gut dysbiosis and cancerous signatures, including stage, localization, or age, with small CRC patient cohorts in this study. The whole genome sequencing is ongoing with the genomic DNA extracted from the paired cancerous tissues. The impact or relevance of MSI with gut dysbiosis in each enrolled CRC patient is ongoing.
- How were the adenomas compared to each other? What classification was used to stage them?
Response: The patients diagnosed with hyperplastic polyp were excluded according to the results of colonoscopy. The collected adenomatous polyps were staged according to the results of pathological examination. Even though the diversity or dissimilarity of gut environment across all patients diagnosed with adenoma was statistically minor, we continuously to enroll more patients with adenoma for addressing the issue.
- How did diabetes etc impact the type and abundance of microbial inhabitants?
Response: All participants diagnosed with DM was excluded in this study. Nevertheless, DM condition is closely related to specific gut dysbiosis in our other study.
- Were surgical patients receiving bowel preparation or were they treated by single shot antibiotics? if yes, please discuss.
Response: In this study, the fecal samples of the enrolled CRC patients were collected prior to the colectomy or rectal resection. Therefore, the impact of bowel preparation or antibiotics on gut microbiota was not discussed.
- The results should be discussed with more similar studies on this topic.
Response: The revised discussion regarding the metabolite-species correlation and its impact on pathogenesis of CRC was provided. Moreover, the potential impact of metabolite-species association that we identified on pathogenesis of CRC was discussed in the revised manuscript (please see Discussion section; page 13, lines 380-384; lines 385-413).